# Variant-Specific Analysis Reveals a Novel Long-Range RNA-RNA Interaction in SARS-CoV-2 Orf1a

**DOI:** 10.3390/ijms231911050

**Published:** 2022-09-21

**Authors:** Matthew Dukeshire, David Schaeper, Pravina Venkatesan, Amirhossein Manzourolajdad

**Affiliations:** 1Department of BioHealth Informatics, Indiana University-Purdue University Indianapolis, Indianapolis, IN 46202, USA; 2Department of Computer Science, Colgate University, Hamilton, NY 13346, USA

**Keywords:** COVID-19, SARS-CoV-2, RNA-RNA interaction, RNA structure, compensatory mutations, viral evolution

## Abstract

Since the start of the COVID-19 pandemic, understanding the pathology of the SARS-CoV-2 RNA virus and its life cycle has been the priority of many researchers. Currently, new variants of the virus have emerged with various levels of pathogenicity and abundance within the human-host population. Although much of viral pathogenicity is attributed to the viral Spike protein’s binding affinity to human lung cells’ ACE2 receptor, comprehensive knowledge on the distinctive features of viral variants that might affect their life cycle and pathogenicity is yet to be attained. Recent in vivo studies into the RNA structure of the SARS-CoV-2 genome have revealed certain long-range RNA-RNA interactions. Using in silico predictions and a large population of SARS-CoV-2 sequences, we observed variant-specific evolutionary changes for certain long-range RRIs. We also found statistical evidence for the existence of one of the thermodynamic-based RRI predictions, namely Comp1, in the Beta variant sequences. A similar test that disregarded sequence variant information did not, however, lead to significant results. When performing population-based analyses, aggregate tests may fail to identify novel interactions due to variant-specific changes. Variant-specific analyses can result in de novo RRI identification.

## 1. Introduction

Severe acute respiratory syndrome coronavirus 2 (SARS-CoV-2) is the cause of the pandemic disease COVID-19 and was first identified as a novel coronavirus by the China CDC [1]. As a result of being infected by SARS-CoV-2, a patient could have a fever, cough, fatigue, sputum production, shortness of breath, and myalgia/arthralgia [2]. As of November 2021, there have been an estimated 260 million cases and 5.2 million deaths [3]. Variants have emerged due to mutations in regions across the globe [4], with four main variants being Alpha, originating in the United Kingdom, Beta, from South Africa, Delta, originally from India and Omicron, which also first emerged in South Africa [5]. Both the Alpha and Beta variants have an estimated increase in transmission of 40–70% due to a mutation in the receptor binding domain [5]. The Alpha variant saw a huge decline in circulation upon the emergence of the Delta variant, although it was noted that the Alpha variant had a slight increase in both transmissibility and severity compared to previous strains. The Beta variant has an additional two mutations that could supplement the virus with a potential resistance to antibodies [5]. The Delta variant also has increased transmissibility that is associated with higher viral load, longer duration of infectiousness, and high reinfection rate, allowing it to become the globally dominant variant over the other two [3]. The Omicron variant has three deletions and one insertion in the Spike protein, increasing its binding affinity to human cell receptor angiotensin-converting enzyme 2 (ACE2), as well as having increased viral replication, viral load, and aid in immune escape after prior infection or vaccination [6].

SARS-CoV-2 is part of the betacoronavirus genus and is a single positive-strand RNA approximately 30 kilobases long [7]. The process of infection begins with the inhalation of airborne particles [8] and then the virus’ spike glycoprotein attaches to ACE2 receptors on the host cells [9]. The virus’ cycle of replication involves continuous and discontinuous transcription processes. RNA-dependent RNA polymerase (RdRp) utilizes the complimentary negative-strand RNA in continuous transcription to generate the genomic RNA and is mediated by a frame shift element (FSE) [10]. For discontinuous transcription, the transcription process is regulated by the transcription regulating sequence-leader (TRS-L) at the 5′ end of the genome and the transcription regulating sequence-body (TRS-B) at the 5′ end of each open reading frame (ORF) by a long-range RNA-RNA interaction that brings the two regions in proximity [11], allowing template switching to terminate transcription at the TRS-L [12]. Thus, the structure of RNA is an important regulator for transcription, translation, and regulation [13].

Long-range RNA-RNA Interactions (RRIs) are crucial for the life cycle of coronaviruses, including betacoronaviruses [14]. One of the roles of genomic long-range RRIs is their involvement in discontinuous transcription, leading to sub-genomic messenger RNA (sgmRNA) production [7]. The interactions bring the TRS-L, located at the 3′ end of the leader sequence, and TRS-B, preceding each viral gene, in proximity, resulting in discontinuous transcription. These interactions have been experimentally verified using a simplified sequencing of the psoralen crosslinked, ligated, and selected hybrids (SPLASH) approach [15]. The long-range RNA-RNA interactions between the TRSs of SARS-CoV-2 are not the only long-range RNA-RNA interactions utilized in the genome. Recent studies used SHAPE-MaP ex vivo [16] and in vivo [17] methods to identify more interactions. A different procedure based on mutational profiling, known as DMS-MaPseq, has also been developed to identify novel interactions in viral RNAs, including SARS-CoV-2 [18]. Another study uses crosslinking of matched RNAs and deep sequencing (COMRADES) to identify long-range RNA-RNA interactions [11]. A few identified long-range interactions in SARS-CoV-2 have been linked with tertiary folding of the genome inside virion [19]. The data obtained from experimental methods such as SHAPE can be used to constrain RNA-RNA interaction prediction. SHAPE directed homology and motif analysis has recently become the preferred method for identifying local RNA-RNA interactions and structures [20]. Often, SHAPE constrained predictions focus on local RNA structure and accessibility; however, it has been shown that long-range RNA-RNA interactions can also be obtained using SHAPE methods [17]. Little work, however, has been conducted to investigate the evolutionary changes of these interactions in the virus’ variants that have emerged over time.

One specific RNA-RNA interaction with a known function in SARS-CoV-2 is the Frame Shifting Element (FSE). The FSE is a *cis*-acting element known to cause a −1 frameshift at the junction of orf1a and orf1ab. It contains a slippery sequence (UUUAAA) where the ribosomal frameshift occurs, a downstream three-stem pseudoknot, and an upstream stem-loop attenuator [10]. The FSE serves to slow the ribosome, which is essential to allow frameshifting to occur. The functional region of the FSE has recently been shown to reside within a 1.5 kb super-structure, labeled as the FSE-arch [11]. Experiments have shown that altering the framshifting mechanism in SARS-CoV-2 has a deleterious effect on viral replication. The FSE has also been shown to be conserved across different coronaviruses [21] and, recently, the stability of the pseudoknot was found to be affected by the tertiary structure of the FSE, as well as take on multiple conformations [22]. Despite the FSE having known features and a known function, the evolution of its features within SARS-CoV-2 variants has not been fully explored.

The rate of evolution changes within SARS-CoV-2 coding and non-coding regions and can be tracked by their mutation rates, which can be different from one another. The genome of SARS-CoV-2 consists of the ORF1a, ORF1b, spike (S), envelope (E), membrane (M) and nucleocapsid (N), where ORF1a and ORF1b encode for 16 non-structural proteins that produce the viral replication and transcription complex [23] and the other genes produce structural proteins by RNA synthesis [24]. Both the membrane and envelope proteins are resilient to high mutation levels. The non-structural ORFs 6, 7a, and 10 have the same properties. On the other hand, the ORFs S, N, ORF3a, ORF8, and ORF1ab are all sources of variation within the genome [25]. Gene ORF8 has had a major deletion in viral strains known as Δ382, which were associated with low symptoms [26]. Across the variants, the S gene has many mutations that have caused changes in virulence. ORF1a began with a few mutations in the first variants, but quickly became a gene with high variation. The rest of the genes follow this same pattern of little initial variation in the variants but increasing over time, but to a much lower degree [6].

One of the major approaches to study the evolution of an RNA structure, however, is through observing co-varying nucleotide changes. Observed changes on either side of an RNA base pair within a population of sequences, i.e., compensatory mutations, can be indicative of how adaptive these intervals are to preserve the stability of the structure of the RNA. Recently, a comparative analysis on long-range RNA-RNA interactions revealed a putative interaction between *Orf8* and *Spike* of SARS-CoV-2 [27]. In this work, we investigate long-range RNA-RNA interactions of SARS-CoV-2 to expand our knowledge of their evolutionary changes within the major viral variants. A selection of experimentally verified long-range RRIs taken from the literature were chosen in this study. We also included several computationally predicted long-range RRIs in our analysis as an exploration into novel long-range RRI and their associated changes.

## 2. Results

Genomic intervals of the RRIs are shown in Table 1. Computational (Comp) RRIs predictions were the result of the IntaRNA [28,29] software on the reference genome (See Materials and Methods). The top 5 hits for each genomic interval were recorded and, subsequently, all interactions were ranked based on energy residuals. The *p*-values reported by IntaRNA all showed significance, and thus were not used to rank computational RRIs (see Methods). The experimentally validated RRIs (Exp) were taken exclusively from [11], where long-range RRIs were reported via COMRADES methodology. All base-pairings for experimental RRIs were obtained from the literature, while the computational RRI structures were obtained from intaRNA. In the eight long-range RRIs there were no pseudoknots or stem-loops; however, these interactions serve as candidates for further exploration. Additionally, the IntaRNA predictions were performed with and without SHAPE data to provide a more constrained prediction. The SHAPE reactivities used were acquired from [17], which performed an in vivo SHAPE mapping of SARS-CoV-2. The results from IntaRNA with the SHAPE constraint showed the identical top hit (Comp2, see Table 1) as the run without SHAPE constraint. In addition to the eight long-range RRIs, the Frame Shifting Element (FSE) was also selected for analysis. The FSE is not a long-range interaction; however, it contains experimentally validated features such as stem-loops and pseudoknots. The structure of the FSE, as well as the labeling of stem-loops, was obtained from [30]. The examination of the FSE was performed to determine if there is any feature-specific evolution across variants.

A total of 32,714 full-genome sequences were downloaded from GISAID.org for the RRI analyses. The data contained sequences from the four variants under study, namely Alpha, Beta, Delta, and Omicron. The sequences originated from 92 different countries, with France being the most prevalent country of origin. A full breakdown of the countries of origin is available in Appendix A. Collection data, as well as the genome-wide mutation rate of each variant, is shown in Table 2. Average pairwise distances between variants were calculated as a measure of dissimilarity. Table 3 demonstrates the differences between the variants of SARS-CoV-2. Delta and Omicron have the largest distance between them and the two closest are Alpha and Beta. Generally, the more recent the variant, the larger the distance between it and the other variants. Despite having the highest average distance, Delta and Omicron still only differ by an average of 87 bases, within a genome of almost 30,000 bases. Within the sample, the genome-wide mutation rate does not follow a chronological pattern, with each variant having different rates of background mutation. Notably, Omicron has the highest background mutation rate and Delta with the lowest rate (See Table 2). These values track, with a daily rate of 2.38 × 10^−6^, or an average of 0.0714, mutations within a genome of SARS-CoV-2 from [31]. Having slightly divergent genomes, the sample was verified as sufficient for our investigation of long-range RRI.

### 2.1. Mutational Analysis of 8 RRI Candidates

To better understand how each SARS-CoV-2 RRI is conserved, regardless of variant, the average mutations per base over all sequences were compared for each of the 8 RRIs. Comp4 had the greatest number of mutations, whereas Exp1 had the least number of mutations (See Table 4). Next, the mutations were categorized as compensatory, meaning they accommodated for the RNA base pairings (A:U, G:C, G:U), and non-compensatory, meaning they disrupted an RNA base pair. Exp2 and Comp1 RRIs showed higher than 90% compensatory mutations, meaning less than 10% of the single nucleotide variations (SNVs) did not comply with the proposed structure. Comp4 had the lowest percentage of non-structural mutations, at just over 12%. This was due to an “identifying mutation” within the Beta sequences in orf3a (Q57H), which did not accommodate the structure, and it is present in nearly all Beta sequences but no other variants. To understand whether certain RRIs experience more/less variation, or if they just fall within highly variable regions of the genome, we compared the per base variation of each RRI to its surrounding region (100 nt up and down stream). As expected, Comp4 showed twice the mutation rate as its surrounding region due to the identifying Beta mutation. Additionally, Exp1, Exp4, and Comp3 showed very little variation relative to their surrounding regions, which corresponds with their lower mutation rates. The mutation analysis for the FSE revealed that it has a mutation rate comparable to the eight long-range RRIs. Unexpectedly, FSE also had more variation per base than the surrounding region, despite it being known that alterations have deleterious effects. This is likely due to the presence of the super-structure it resides in, as the surrounding bases would also be under selective pressures. Finally, the ratio of compensatory mutations for the FSE was quite high as expected, providing further evidence that the FSE structure is under structural constraints.

A covariation analysis did not reveal any significantly co-varying base-pairs within any of the 8 RRI candidates; however, the survival plots of exp4 and comp2 came close to having a co-varying nucleotide pair that passes the 0.05 significance threshold (See Figure 1). Additionally, the FSE showed no significant covariation. R-scape reports the power and covariation scores that it uses to compute E-values. To know if an alignment has sufficient variation, one must look at the power value which ranges from 0 to 1. A low power indicates the covariation analysis is inconclusive. High power (>0.6) and no covariation can serve as evidence against a conserved structure. Despite no significance being found, it does not mean that the interactions do not exist, as Exp1–4 have already proven to exist by experimental means, but rather that there may not be enough variation to pass the statistical test or certain interactions may be well conserved in specific variants and thus are hidden in the population. For Exp4, specifically, the power level was high, and the amount of covariation was near significant. This is contrasted by Comp2, which had a low power level and yet was close to significance due to just a few covarying bases. The FSE showed moderate power levels for most base-pairs, with a few having low power.

### 2.2. Variant-Specific Analysis

Following the initial analysis of the 8 RRI regions, the sequences were then grouped by their corresponding variants to investigate any trends specific to a variant or group of variants. The calculations performed were identical to the initial analysis. First, the variation relative to the surrounding region showed similar overall trends compared to the first analysis, as expected (see Table 5). However, it was observed that there are differences between the variants for certain RRI regions. For example, Comp1 had lower variation compared to the surrounding region for all variants except Delta (number1 vs. other numbers), and Exp1 had lower relative variation in Delta and Omicron but not in Alpha and Beta (similarly give numbers). Additionally, Comp4 had much less relative variation in Omicron, while it was much higher for the other three variants (See Table 5). However, Comp2 and Comp4 each saw two variants with a higher variation relative to their surrounding regions. Finally, the mutations were categorized as compensatory mutations or non-compensatory mutations. Most RRIs show discrepancies across variants in terms of % compensatory mutations. For instance, in the RRI Comp4 where the identifying mutation was seen in the Beta variant, only 3% of Beta mutations accommodate the structure while 90% of Alpha mutations do. Additionally, it was observed that Exp2 and Exp4 both showed an increase in the ratio of mutations which accommodated the structure in Alpha to Omicron. Finally, the variant-specific analysis of the FSE revealed a stark difference between variants with respect to both relative variation and compensatory mutations. The results of our data are suggestive of the fact that long-range RRIs may be undergoing different evolutionary pressures in different SARS-CoV-2 variants.

To determine whether any of the RRI candidates were evolving in a variant-specific manner, sequences from each variant were separated and an identical R-scape analysis was performed. The results showed that a majority of the RRIs still show no significance for any variant; however, the Exp4 interval of Beta sequences, namely Beta Exp4, and Beta Comp1, both showed significantly covarying base pairs (Figure 2a,b), despite having low power. The Omicron Exp4 and Beta Comp3 analysis also showed some covariation; however, neither were below the significance cutoff. Conversely, Comp1 showed little covariation in all variants other than Beta. Again, most of the analysis resulted in low power values, indicating a lack of variation in many RRIs for most variants.

### 2.3. SHAPE Constraint Results

Once the predicted RRIs were analyzed, IntaRNA was used again, this time with added SHAPE constraint data. These results were ranked (See Methods) and the top 5 selected for further variant-specific analysis. Interestingly, the number one ranked prediction is the same region as Comp2. Because of this, the next four were selected (Table 6) for covariation analysis through R-scape. All these computational RRIs with SHAPE data did not have any significant covariation. Omicron Shape1 and Delta Shape3 came close (Figure 3); however, the rest did not. In the case of Omicron Shape1, it had low power and Delta Shape3 had high power for some base pairs. The rest were mostly low power, again.

## 3. Discussion

Long-range RRIs have are critical to the life cycle of coronaviruses in general. Identification of these high-order RNA-RNA interacting regions can enhance our knowledge of the viral life cycle. In addition, SARS-CoV-2 is a persistent virus which is constantly evolving into new variants. Knowledge on the evolutionary changes of long-range RRIs can also aid more effective characterizations of different variants and potentially help predict emerging ones. In this work, we focused on monitoring meaningful changes in long-range RRIs within the population of gathered sequences. For this purpose, we selected a total of eight long-range RRIs from both the literature (four RRIs) and computational predictions (four RRIs). Our computational predictions were a result of an in silico fragment-based approach that deployed IntaRNA software. We then selected four candidates based on the thermodynamic stability of predictions. The purpose of including computational predictions was to further explore the space of possible long-range RRIs and possibly predict emerging RRIs that were not yet experimentally validated. We first investigated RRI co-evolution by investigating mutational patterns within the population of gathered sequences. Subsequently, to investigate any variant-specific RRI changes, we investigated the same mutational patterns within the population of sequences belonging to a specific variant. Our study included Alpha, Beta, Delta, and Omicron variants.

Genomic coordinates of the four long-range RRIs in this study are shown in Table 1. The coordinates of both experimental (Exp) and computational (Comp) RRIs include a variety of genes and regions on the genomic RNA. Data consisting of over 32,000 sequences obtained from GSAIDwas investigated for assessing evolutionary changes. To explore RRI structural changes, we looked at distinctive features such as the number of compensatory mutations, conservation (relative variation), and significance of co-varying mutations for each RRI. Table 4 summarizes the statistics for sequence conservation and co-evolution. We can generally see that mutation rates are heterogeneous across RRI regions, reconfirming different evolutionary rates and constraints on different regions of the genomic RNA. Certain RRIs such as Exp1 and Exp4 showed relatively high conservation, while other RRIs such as Exp2 and the predicted RRI Comp1 contained high numbers of compensatory mutations. Statistical analyses performed via R-scape, however, did not indicate any significantly co-varying mutation for any of the eight RRIs, as all *p*-values were higher than the 0.05 threshold. These values are not corrected for multiple testing and the real significance values should be even less than individually obtained *p*-values. Computational predictions were repeated, using SHAPE data as a constraint, and no significance was found in the top hits when using SHAPE. Although imputing SHAPE data for RRI prediction is becoming more popular, especially for local structure prediction, it should be used with caution when mapping long-range RRIs. Given our data and analysis, there is no evidence that the two regions of any investigated RRI are co-evolving within the population of *sequences*. The low significance of results were likely due to two factors. First, it could have been that not enough mutation had occurred within the tested population of sequences, which could be due to other constraints on these genomic intervals [33]. Rivas et al. mention that a lack of significant E-values does not necessarily argue against a conserved structure; an alignment could simply have too little variation for the co-varying bases to be deemed significant [34]. Second, it could have been that different sub-categories or variants of SARS-CoV-2 virus are evolving differently, causing the overall aggregate insignificant results. For the latter, we chose to further investigate variant-specific compensatory mutations.

The SARS-CoV-2 variant had different mutation rates and evolutionary patterns. On a genomic level, Table 2 shows the average pairwise distance between the variants. Omicron, being the more recent variant, had the highest sequence divergence from other variants. Omicron is also observed to have a higher mutation rate than the other three variants (See Table 3). Delta, on the other hand, had the lowest mutation rate. Table 2 suggests that Alpha and Beta variants are closer to each other than to other variants.

Mutational rates and patterns were different across the RRIs, as well as across variants. Variant-specific analysis shown in Table 5 suggests heterogeneity of sequence conservation, as well as that of compensatory mutation. Long-range RRIs Exp1 and Comp4 showed the highest dispersion of compensatory mutation rates across variants, where dispersion is quantified by the standard deviation in average observed compensatory mutation rates across variants. Here, we will briefly discuss some of the RRIs for which we observed extreme values.

### 3.1. FSE Mutations Were Not Consistent across Variants

Overall, the somewhat high relative variation seen for the FSE across all sequences was expected, due to the surrounding super-structure also being under selection. The ratio of compensatory mutations was quite high, indicating it is under selection in the population. For the variant-specific analysis, Alpha and Delta had a higher ratio of compensatory mutations compared to Beta and Omicron. The FSE in both the Alpha and Omicron variants showed a higher mutation rate than the surrounding region. However, the ratio of compensatory mutations in Alpha was much higher than that of Omicron. Additionally, the mutation rate for the FSE in Beta and Delta was lower than the surrounding region. Despite this, Beta and Delta show a similar difference in compensatory mutations as Alpha and Omicron. To better understand the feature-specific variation within each variant, mutations were grouped by feature. The slippery sequence had no mutations in any variant, which was expected because the specific bases UUUAAA are required for ribosomal frameshifting to occur in SARS-CoV. The upstream stem-loop attenuator showed only a few mutations, none of which were consistent across variants. Finally, analysis of features within the FSE revealed some differences between variants. Both Beta and Omicron had most mutations within unpaired bases of the FSE, as expected. The Alpha variant, however, had most of the mutations reside in the pseudoknot of the FSE, all of which accommodated the structure. Overall, stem-loop 1 and stem-loop 3 contained very few mutations in all variants except for Delta. Delta contained nearly 75% of the FSE mutations within stem-loop 1, most of which accommodate the structure, and almost no mutations within the pseudoknot. The differences in mutations for specific features of the FSE indicate variant-specific selection. (NO mutations at all in slippery sequence upstream of FSE.) Additionally, high relative variation makes sense because FSE resides in FSE-arch super structure which is also conserved. Additionally, the attenuator hairpin upstream of the slippery site experienced mutations at about half the rate of the FSE, showing 5.50E-05 mutations per base per sequence. The results suggest the variant-specific evolution of the FSE and further analysis of the function and structure of this feature should consider variant specificity.

### 3.2. Exp1 Is Variant Specificity for Orf1ab-Orf1ab RRI (Exp1)

The Delta variant had the highest percentage of compensatory mutations in Exp1 (0.83) compared to those of other variants. Omicron variant had the lowest percentage (0.33), while showing lower sequence conservation in this region. While the average percentage of compensatory mutation was only 0.6538 (Table 4), variant-specific tests revealed divergent behavior. It is suggested that Exp1 RRI is more structurally conserved in the Delta variant than other ones. In addition, heterogeneous sequence conservation seen in Table 5 further suggests that this region is undergoing different evolutionary changes in different variants, where it is much less conserved in the Omicron variant than others. Given that the average percentage of compensatory mutations had the fourth highest dispersion amongst the eight RRIs, we speculate that the Exp1 long-range interaction may be undergoing variant-specific evolutionary changes over time. The covariation analysis provided no further insight, as the low power values indicate there is not enough variation for the statistical test.

### 3.3. Another Orf1ab-Orf1ab RRI (Exp2) Is Consistent across Variants

The Exp2 long-range RRI had the highest percentage of compensatory interactions observed amongst the eight studied RRIs, with the percentage of compensatory mutations being 0.9472 (See Table 4). The two intervals of Exp2 both occur in Orf1ab and, interestingly, contain more mutations than their surrounding regions, where relative variation is around 1.2 (See Table 4). Variant-specific tests do not suggest any heterogeneity across variants as the percentage of compensatory mutations are above 0.5 for all variants and the dispersion of values is the lowest among the eight RRIs (See Table 5). Overall, our results suggest that Exp2 is consistently present in all four SARS-CoV-2 variants and does not undergo any dramatic changes. This is also supported by the covariation analysis, which shows low variation for this RRI in all variants. The biological relevance of this interaction and Exp1 has not been explored, though it was proposed that they play roles in discontinuous transcription and/or replication [11].

### 3.4. Orf1a-Orf1a RRI (Exp4) Is Consistent across Variants with Significant Covariation

The Exp4 long-range RRI had a percentage of compensatory mutations ranging from 0.5 in Beta to 0.85 in Omicron, with the average rate being 72.5%, ranking it as third highest among the eight RRIs (see Table 4 and Table 5). Additionally, the relative mutation rates for all variants were low, indicating conservation of this region relative to flanking regions. Both interacting regions of Exp4 lie within Orf1a, so the consistent low relative variation was unexpected. The 5′ region of this interaction is also reported to interact with two other regions of Orf1a, one being the Exp3 interaction and the other being 2 kb upstream [11]. These three interactions likely play a role in the discontinuous transcription or viral replication by bringing elements into physical proximity. Exp4 was found to be consistent across variants, as the low dispersion value suggests (see Table 5). Thus, similar to Exp2, it was concluded that Exp4 likely is not undergoing any dramatic changes. The covariation analysis did report significant covariation in the beta variant; however, covariation was also present in Alpha and Omicron despite it not being significant. Overall, it was concluded that Exp4 likely is not showing variant-specific conservation, but future analysis after more mutation accumulation is needed due to Exp4 showing near significance for the analysis of all sequences (see Figure 1).

### 3.5. Predicted Orf1a-Orf1a RRI (Comp1) Shows Evidence Suggestive of Variant Specificity

The computationally-predicted RRI Comp1 was observed to have a high percentage of compensatory mutations. Comp1′s interacting regions have similar statistics to Exp2 in that they contain both high compensatory mutations and higher mutations than surrounding regions (See Table 4). Both intervals of Comp1 are in Orf1a. Table 5 suggests a high dispersion of average compensatory mutation values across variants, with Omicron containing less compensatory mutations than the other variants in this region. Given the ranking of this region in our computational analysis and the high percentage of observed compensatory mutations in this interval, we speculate that Comp1 may be a novel long-range RRI. Dispersion values, however, are not extreme enough to suggest variant specificity for this putative RRI. The likelihood of variant specificity is especially low, since the variant with lower compensatory mutation, namely Omicron, has a more divergent sequence. Given that our computational predictions were based only the reference sequence, it is expected that the Omicron and Delta variants would have diverging percentages of compensatory mutations and base-pairing mismatches. Additionally, the covariation analysis showed significant covariation for Beta Comp1. The lack of significance in other variants does not mean the interaction is specific to Beta, as the low power values reported by R-scape indicate not enough variation has occurred for the test to be relevant. Overall, the high percentage of compensatory mutations along with the lower relative mutation rate and significant covariation in Beta serve as evidence for the existence of this RRI.

### 3.6. Orf1a-Orf1ab RRI (Comp4) Shows Statistical Evidence of Variant Specificity

The Comp4 prediction occurs between Orf1a and Orf3a. This is an interesting prediction. At first glance, there seems to be high sequence variation (low sequence conservation) in this region and there is a low percentage of compensatory mutations associated with it (See Table 4). When we break the data into variants, however, we observed the highest dispersion amongst variants, SD = 0.3566861. Sequences in the Alpha variant contained as high as 0.9 compensatory mutations, while this value was dramatically lower in the Beta variant, dropping to 0.03 (See Table 5). This is rather surprising since both Alpha and Beta variants were observed to have relatively high mutations in Comp4 intervals and in regions surrounding it. The Beta variant’s high number of mutations are due to an identifying mutation in the interacting region, not present in the other variants. Unlike Alpha, the mutations occurring in the Beta variants do not accommodate for the long-range RRI prediction. Figure 2a,b show the trend of Comp4 mutation over time. As we can see, mutations occurring from the summer of 2021 to winter of 2021 in the Beta variant, shown in green, contain a much lower percentage of compensatory mutations. No significant covariation was observed for any of the variants; however, like many of the other RRIs, the covariation analysis reported low power values, indicating an overall lack of variation in all variants other than Beta, due to its identifying mutation.

## 4. Materials and Methods

### 4.1. Long-Range RRI Selection

Four experimentally verified long-range interactions were chosen to be included in the analysis. These ranges were chosen from the COMRADES experiment performed by Ziv et al. [11]. Of the eight reported long-range interactions, those which were located near a known conserved RNA structure were prioritized, assuming that their likelihood for being structurally functional would be higher [35]. Additionally, long-range RRIs were predicted using IntaRNA 3.2.0 (University of Freiburg Breisgau, Germany) [28,29]. The reference genome for SARS-CoV-2, NC_045512.2, was used to make the interaction prediction, utilizing slices of 200 nt and 500 nt windows with an overlap of 50 nt of the reference genome. The focus of this study is on long range interactions, so any predicted RRI where the beginning of 5′ end and beginning of the 3′ were less than 2000 nt apart were filtered out. Additionally, since the 5′ and 3′ UTR have been studied in detail and bind to many locations in the genome, any interactions predicted in these regions were filtered out as well in order to focus on regions in the SARS-CoV-2 genome that are less studied in detail. In order to rank the results, the residual values from performing generalized linear regression on the length of the 5′ end of the interaction, plus the length of the 3′ end of the interaction normalized by the interaction energy value, were used. The most negative residual values were chosen as the highest ranks because that shows the RRI has a much lower energy level than would be expected based upon the regression. This method was used instead of the *p*-values, because they were all significant. The overall top hits were the cumulative top hits across all runs (see Table 1). To provide more context to make a more constrained prediction, IntaRNA was again used with SHAPE data from [17], following the same procedure for filtering and ranking the results.

### 4.2. Sequence Collection

Data were collected from the GISAID database (www.GISAID.org, accessed on 20 May 2022). Only sequences labeled as the variants Alpha, Beta, Delta, and Omicron were used. Additionally, the sequences were required to be complete with high coverage, along with a collection date. All the available Beta (4237) and Omicron (8457) sequences at the time of collection were selected to not overly unbalance the dataset while still collecting a large sample set; 10,000 sequences of Alpha and Beta were sampled. The sequences used have different ranges of submission dates for each variant, as shown in Table 2. Additionally, the sequences used come from 92 different countries, with about half of the data originating from France, Italy, USA, or Mexico. The sampled genomes were then aligned using MAFFT [36,37]. Once the sequences were collected and aligned, preprocessing was carried out such that the interacting regions from the chosen experimentally determined RRIs from [11] and the top computationally predicted RRIs were selected and collated into Stockholm formatted files, first grouped only by the RRI, and then by each variant within the RRI group for further analysis. Interaction sequences that contained gaps were permitted in our analysis and R-scape also has methods to handle them, so they were not filtered out.

### 4.3. Mutation Calculations

The mutations were calculated by using the Stockholm files generated, and where the bases in the sampled sequences differed from the reference genome it was counted as a mutation. The count was then normalized by the length of the interacting sequence. % Compensatory refers to the percentage of mutations which do not disrupt the structure, specifically in the case of alternative base pairing such as GU [38]. It is derived by dividing the number of compensatory mutations by all mutations within an RRI. Relative variation shows the number of mutations in an RRI per base, divided by that of the surrounding region (100 nt up and down stream); a value higher than 1 would indicate more variation within the RRI compared to the context.

### 4.4. Covariation Analysis

Using R-scape 2.0.0 (Harvard University Cambridge, Massachusetts USA) [33,34,39] and a proposed structure calculated through the Bifold tool from RNAstructure 6.4 (University of Rochester Medical Center Rochester, New York, NY, USA) [40,41], a covariation analysis was performed for each RRI across all sequences, with an E-value significance level of 0.05. This was carried out to assess co-evolving regions of the experimental RRIs and to validate the existence of the computationally predicted RRIs. Additionally, this analysis was performed to validate the results of the structural mutation calculations. A similar analysis was subsequently performed for each RRI but separated by variant to elucidate any variant-specific co-evolution that may be lost when all variants are grouped together. For all R-scape analyses, the recommended default parameters were used due to the two-set test not being sensitive enough for our data. Survival Plots were generated for each R-scape run. The ‘--gapthresh’ option was used to omit sequences with gaps.

## 5. Conclusions

Our population-based analyses suggested variant specificity for certain RRIs, despite the covariation analysis having lower power than many of the RRIs, is the most powerful way to identify significantly co-varying nucleotides within. One of the computationally predicted RRIs, Comp1, was found to have statistically significant covariation in the Beta variant. Interestingly, this significant covariation was only present when the sequences were not aggregated (Table 4, Figure 1), but rather separated into variant-specific sets, and only seen within the beta variant, as seen in Table 5 and Figure 2b. This demonstrates the variant specificity of the interaction, located within Orf1a. It was shown that an experimentally verified long-range RRI, here Exp1, can have variant-specific behavior as well, see Table 5 and Figure 2a. Thus, our results suggest that when performing population-based analyses, aggregate tests may not be sufficient to identify novel RRIs. Certain RRIs are evolving differently in different variants, and it is critical to study their changes in sub-populations of SARS-CoV-2 sequences. The Comp1 interaction was not detected in the entire population of sequences (see Table 4); however, when sequences were grouped by variant it was detected as significantly covarying (see Table 5 and Figure 2). Additionally, evidence was found for variant-specific evolution of the FSE, which has been proposed as a possible drug target. Variant specificity should be considered when targeting this structure for drug treatment of SARS-CoV-2. Furthermore, SHAPE constrained predictions revealed no significant covariation and should likely be used with caution when mapping long-range RRIs. As some of the RRIs show variant-specific behavior, it is demonstrated that one component of evolution of the SARS-CoV-2 virus is how it interacts within itself, alongside the interactions the virus has with the host. Experimental procedures must be carried out to further understand the function of the novel Orf1a RRI and the mechanism by which it acts. Future studies, conducted after more mutations accumulate in SARS-CoV-2 variants, will provide further insights into the variant-specific evolution of these RRIs.

## Figures and Tables

**Figure 1 ijms-23-11050-f001:**
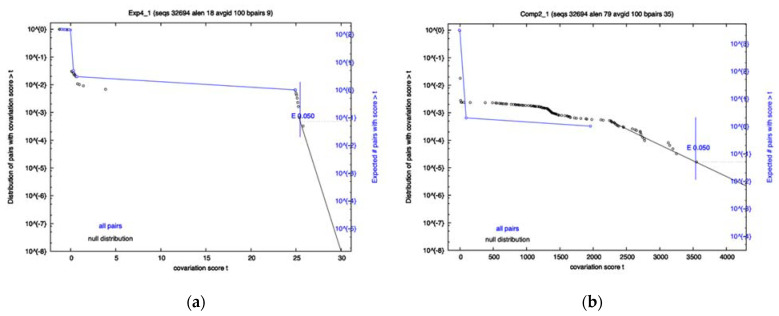
Survival Plots of two of the 8 RRI candidate regions. Each plot has two lines: Black indicates a null distribution; Blue is all base pairs in the interaction. Exp4 is shown on Panel (**a**), with Comp2 on Panel (**b**). Significance cutoff used was 0.05.

**Figure 2 ijms-23-11050-f002:**
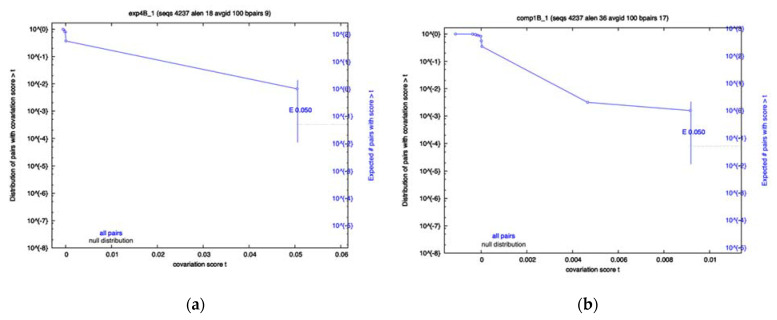
Survival plots from the variant-specific covariation analysis. Displayed are the most significant results. Beta showed significant covariation for Exp4 panel (**a**) and Comp1 panel (**b**). Significance cutoff E-value used was 0.05.

**Figure 3 ijms-23-11050-f003:**
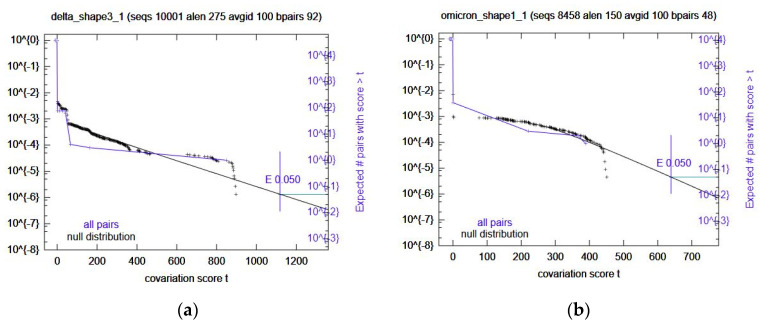
Survival plots from the variant-specific covariation analysis. Displayed are the most significant results. Omicron Shape1 is shown in panel (**a**) Delta Shape 3 is shown in panel (**b**). Significance cutoff E-value used was 0.05.

**Table 1 ijms-23-11050-t001:** List of experimentally reported and computationally predicted long-range RRIs used in the analysis. The first four are reported from a COMRADES experiment, the last four were predicted using IntaRNA, both with and without SHAPE reactivities as a constraint.

NAME	5′ START	5′ END	5′ GENE	3′START	3′END	3′ GENE	ENERGY	# NEARBY LOCAL STRUCTURES/ RESIDUAL
EXP1	15,424	15,433	orf1ab	17,442	17,452	orf1ab	−7.4	6
EXP2	18,541	18,554	orf1ab	21,107	21,119	orf1ab	−12.3	1
EXP3	2070	2081	orf1a	5688	5699	orf1a	−20.5	4
EXP4	5652	5661	orf1a	9076	9084	orf1a	−11	2
COMP1	1394	1411	orf1a	10,951	10,968	orf1a	−21.55	−6.94
COMP2	11,912	11,950	orf1a	18,905	18,944	orf1ab	−21.88	−8.97
COMP3	3047	3073	orf1a	28,809	28,836	N	−20.67	−8.74
COMP4	3177	3194	orf1a	25,546	25,563	orf3a	−20	−8.89
FSE	13,462	-	orf1a	-	13,545	orf1ab	−27.2	FUNCTIONAL IMPORTANCE

**Table 2 ijms-23-11050-t002:** Dates of sequences from each variant as well as calculated background mutation rate. Previous reports are 2.38E-06 per base per day.

Variant	Oldest Sequence	Most Recent Sequence	Background/Genome-Wide Mutation Rate (per Base, per Day)
**Omicron**	8 November 2021	17 May 2022	2.51 × 10^−6^
**Delta**	22 September 2020	15 August 2021	1.96 × 10^−6^
**Beta**	18 February 2020	5 November 2021	2.03 × 10^−6^
**Alpha**	1 December 2020	20 April 2021	2.43 × 10^−6^

**Table 3 ijms-23-11050-t003:** Pairwise distances between variants (mean number of differing bases for all sequences). Calculated from the GISAID data using the MEGA software [32]. Number of sequences: Omicron-8457, Delta-10000, Beta-4237, Alpha-10000.

Pairwise Distances				
**Omicron**				
**Delta**	86.88513			
**Beta**	79.98906	56.01551		
**Alpha**	78.80249	60.95150	49.47034	
	**Omicron**	**Delta**	**Beta**	**Alpha**

**Table 4 ijms-23-11050-t004:** Analysis of 8 RRI candidates across 32,714 SARS-CoV-2 genome sequences. Mutations per sequence were calculated and normalized by interacting sequence length.

Name	Mutations per Base per Sequence	Relative Variation	Ratio of Compensatory Mutations
Exp1	4.29 × 10^5^	0.016	0.654
Exp2	3.96 × 10^4^	1.209	0.947
Exp3	1.28 × 10^4^	0.307	0.440
Exp4	6.31 × 10^5^	0.059	0.725
Comp1	1.51 × 10^4^	1.510	0.924
Comp2	1.08 × 10^4^	1.440	0.521
Comp3	9.36 × 10^5^	0.009	0.315
Comp4	7.62 × 10^3^	2.203	0.125
FSE	1.35 × 10^4^	1.383	0.882

**Table 5 ijms-23-11050-t005:** Variant-specific analysis of the 8 RRI candidates. Relative variation is normalized by the number of sequences (value of 1 indicates an equal per base variation in the surrounding region, 100nt in each direction). E-value reports the significance value of the R-scape covariation analysis, with the level of power categorized as low if it is reported to be <10% (power is fraction of base pairs expected to covary). Dispersion represents the standard deviation of % compensatory mutations for all variants.

Name	Variant	Variation Relative to Context	Ratio of Compensatory Mutations	E-Value of Significantly Covarying Base Pairs (Low/High Power)	Dispersion
Exp1	Alpha	0.409	0.500	NA (low)	0.208
	Beta	0.935	0.571	NA (low)	
	Delta	0.011	0.833	NA (low)	
	Omicron	0.006	0.333	NA (high)	
Exp2	Alpha	0.180	0.684	NA (low)	0.140
	Beta	3.916	0.993	NA (low)	
	Delta	1.919	0.943	NA (low)	
	Omicron	0.281	0.800	NA (high)	
Exp3	Alpha	0.141	0.714	NA (low)	0.200
	Beta	1.070	0.250	NA (low)	
	Delta	0.499	0.377	NA (low)	
	Omicron	0.160	0.545	NA (high)	
**Exp4**	Alpha	0.344	0.600	NA (low)	**0.155**
	**Beta**	**0.892**	**0.500**	**0.05 (low)**	
	Delta	0.025	0.750	NA (high)	
	Omicron	0.765	0.846	NA (high)	
**Comp1**	Alpha	0.277	0.714	NA (high)	**0.256**
	**Beta**	**0.316**	**0.750**	**0.05 (low)**	
	Delta	2.377	0.978	NA (low)	
	Omicron	0.667	0.357	NA (high)	
Comp2	Alpha	0.673	0.722	NA (low)	0.229
	Beta	0.505	0.909	NA (low)	
	Delta	1.982	0.360	NA (low)	
	Omicron	1.932	0.708	NA (low)	
Comp3	Alpha	0.007	0.208	NA (low)	0.157
	Beta	0.016	0.400	NA (low)	
	Delta	0.016	0.309	NA (low)	
	Omicron	0.004	0.577	NA (high)	
Comp4	Alpha	3.307	0.898	NA (low)	0.357
	Beta	6.028	0.033	NA (low)	
	Delta	1.329	0.500	NA (low)	
	Omicron	0.051	0.424	NA (low)	
FSE	Alpha	1.626	0.968	NA (low)	0.208
	Beta	0.668	0.571	NA (low)	
	Delta	0.982	0.887	NA (low)	
	Omicron	1.383	0.571	NA (low)	

**Table 6 ijms-23-11050-t006:** List of computationally predicted RRIs with SHAPE data used in the analysis.

NAME	5′ START	5′ END	5′ GENE	3′START	3′END	3′ GENE	ENERGY	RESIDUAL
SHAPE1	15,893	15,970	orf1ab	28,383	28,454	N	−25.77	−8.40
SHAPE2	20,068	20,193	orf1ab	22,327	22,433	S	−27.22	−8.32
SHAPE3	9077	9222	orf1a	12,748	12,876	orf1a	−27.86	−8.19
SHAPE4	5367	5402	orf1a	24,114	24,157	S	−24.21	−8.14

## Data Availability

Available upon request.

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
