# Peer review of "Variant-Specific Analysis Reveals a Novel Long-Range RNA-RNA Interaction in SARS-CoV-2 Orf1a"

_ijms, 2022, doi:10.3390/ijms231911050_

Round 1

Reviewer 1 Report

The current manuscript seems to be a followup of an earlier paper published in late 2020 (10.1016/j.molcel.2020.11.004). Authors have tried to expand the analysis with currently dominating variants of concern. And tried to match the conservancy of these interactions. While the original paper was important as a pioneering study there have been a lot of development in the field since then and has enhanced our understanding of intragenomic RRIs of SARS-CoV-2.

After the in vivo structure of genomic RNA was experimentally revealed there were some substructures found (10.1016/j.molcel.2020.12.041) which were also found in earlier paper. Features like stem loops, pseudoknots, and ribosome frameshift signal were discovered in consecutive studies (10.1128/mmbr.00057-21).

Now the trending research in the field is focussed on Shape-guided RNA structure homology (10.1038/s41467-022-29398-y, 10.1016/j.gde.2021.11.007, 10.1128/mmbr.00057-21 and 10.21203/rs.3.rs-1160075/v1).

While authors have followed a critical paper and tried replicating results with variants, they have completely ignored the depth of information that followed regarding RRIs of SARS-CoV-2. Even there are a few hypotheses that abundant miRNAs and snRNAs coded by host genome have selected out certain features in the viral genome. This study is too superficial to be of a significance to scientific community.

Authors need to talk about specific features and their evolution among different variants. The paper in its current form is too rudimentary.

Recommendation: Major revisions

Author Response

Hello Reviewer,

Thank you very much for your detailed revisions. We have incorporated much of what you've suggested within our revised manuscript.

"After the in vivo structure of genomic RNA was experimentally revealed there were some substructures found (10.1016/j.molcel.2020.12.041) which were also found in earlier paper. Features like stem loops, pseudoknots, and ribosome frameshift signal were discovered in consecutive studies (10.1128/mmbr.00057-21)."

We incorporated the publications you mentioned which were related to long-range RNA-RNA interactions into the introduction. Additionally, we've included a variant specific analysis of the well-known Frame Shifting Element for SARS-CoV-2 and it's specific features (pseudoknot, stem-loops).

"Now the trending research in the field is focussed on Shape-guided RNA structure homology (10.1038/s41467-022-29398-y, 10.1016/j.gde.2021.11.007, 10.1128/mmbr.00057-21 and 10.21203/rs.3.rs-1160075/v1)."

We also were aware of the use of SHAPE reactivities for constraining RNA-RNA interaction predictions, so we've run our computational predictions again with SHAPE reactivities and included them in the results section, as well as mentioning the drawbacks and takeaways.

Thank you again for your time and effort in providing revisions. We look forward to your response.

Reviewer 2 Report

The manuscript ID ijms-1873228 "Variant Specific Analysis Reveals a Novel Long-Range RNA-2 RNA Interaction in SARS-CoV-2 Orf1a" is an interesting study. Since the COVID-19 pandemic, understanding the pathology of the SARS-CoV-2 RNA virus and its life cycle has been the priority of many researchers. Currently, new variants of the virus have emerged with varying levels of pathogenicity and abundance within the human-host population. Although much of viral pathogenicity is attributed to the viral Spike protein’s binding affinity to the human lung cell’s ACE2 receptors, comprehensive knowledge about all the distinctive features of viral variants that might affect their life cycle and pathogenicity is yet to be attained. Recent in-vivo studies into the RNA structure of the SARS-CoV-2 genome have revealed long-range RNA-RNA interactions. Using in-silico predictions and a large population of SARS-CoV-2 sequences, we observed variant-specific evolutionary changes for certain long-range RRIs. We also found statistical evidence for the existence of one of the thermodynamic-based RRI predictions, namely Comp1, in the Beta variant sequences. A similar test that disregarded sequence variant information did not, however, lead to significant results. When performing population-based analysis aggregate tests may miss identifying novel interactions due to variant-specific evolution. Variant-specific analyses can result in de novo RRI identification.

 1)      However, the following changes are required to improve the quality of the manuscript

 2)      The authors stated, "The binding energy was normalized by a function of the length of the interaction, and the results were ranked by this value" How did the author's calculated binding energy? describe in detail with proper citation

 3)      The authors stated, "any interactions within the 3’ or 5’ UTR were thrown out as well" what were the unfavorable interactions observed and removed? describe in detail

 4)      How about pre-processing of raw FASTA seq? from the database collection

 5)      The authors stated, "the sequences used 368 come from 92 different countries" what is the ratio of selecting 92 countries? what are the countries or any reference links?

 6)      How are the mutations calculated from RRI? describe in detail with proper citation

 7)      Not clear??? "[Please explain in a lines that sequences were taken only from COMRADES paper] [also touch on the residual ranking and refer the reader to Materials and Methods]

 8)      Missing full form "EXP1 and COMP1" in the main text and table

 9)      What is RESIDUAL RANKING? how did the authors perform this calculation?

 10)   The conclusion needs to be re-write because it lacks meaningful information from the study.

An overall observation of the good study needs critical revisions on methodology, results, and conclusion sections.

Author Response

Hello Reviewer,

Thank you for your detailed comments and revisions, we've incorporated much of what you've suggested into our revised manuscript. Here is a response, point by point according to your revisions.

2) We've included the details about the energy calculation in the methods section. The binding energy is calculated by the IntaRNA software using thermodynamic models. The citation is available in the manuscript.

3) The 3' and 5' UTR regions tend to always show up as hits in computational predictions because of the TRS elements involved in transcription. Many of these interactions are already known and are conserved across many viruses. This is now included in the introduction and methods section.

4) Sequences downloaded from the GISAID Database come in the form of FASTA, and MAFFT was used to align all sequences to the SARS-CoV-2 reference genome. This is now described in detail in the methods section.

5) We have now included a supplementary table showing the number of sequences from each country. The sequences were not hand selected, but rather downloaded in batch from GISAID. The obtained sequences were the most recent of their respective variant.

6) Mutations are calculated in RRI regions by counting the SNPs in each sequence by comparing to the reference genome (not including gaps or erroneous bases such as N). This is also mentioned in the methods section.

7&8) Fixed these formatting mistakes.

9) We have included a detailed description of how this was performed in the methods section. We simply applied a generalized linear regression model to the normalized energy values obtained from intaRNA.

10) We have done major revisions to the Discussion section and some additions to the conclusions section to wrap up the paper.

Thank you again for you time in providing detailed and structured revisions.

Round 2

Reviewer 1 Report

All of my comments from round one have been satisfactorily met

Reviewer 2 Report

The authors addressed all the comments raised by the reviewer. Accept in present form.